# Towards the Development of an Intervention to Address Social Determinants of Non-Communicable Disease in Kerala, India: A Mixed Methods Study

**DOI:** 10.3390/ijerph17228636

**Published:** 2020-11-20

**Authors:** Martin Webber, Jacques Joubert, Meredith Fendt-Newlin, Saju Madavanakadu Devassy, Lorane Scaria, Anuja Maria Benny, Lynette Joubert

**Affiliations:** 1International Centre for Mental Health Social Research, Department of Social Policy and Social Work, University of York, York YO10 5DD, UK; newlin.meredith@gmail.com; 2Department of Neurology, St Vincent’s Hospital Melbourne, VIC 3065, Australia; jacques.joubert@me.com; 3Department of Social Work, Rajagiri College of Social Sciences, Cochin, Kerala 683 104, India; saju@rajagiri.edu (S.M.D.); loraneskaria222@gmail.com (L.S.); anuja@rajagiri.edu (A.M.B.); 4Department of Social Work, Melbourne School of Health Sciences, University of Melbourne, Melbourne, VIC 3010, Australia; ljoubert@unimelb.edu.au

**Keywords:** cardiovascular disease, hypertension, diabetes, social risk, social intervention, social capital, social support, theory of change

## Abstract

In India, cardiovascular disease (CVD), with hypertension as its foremost risk factor, has the highest prevalence rate of non-communicable diseases (NCDs) and a rising mortality. Previous research has found a clustering of behavioural and social risks pertaining to NCDs, though the latter are infrequently addressed in public health interventions in India. This paper reaches toward the development of a social intervention to address social determinants of NCD relating to hypertension and diabetes. We used Theory of Change (ToC) as a theoretical approach to programme design. Mixed methods were used, including qualitative interviews with community members (*n* = 20), Accredited Social Health Activists (*n* = 6) and health professionals (*n* = 8), and a stakeholder workshop (*n* = 5 participants). The recruitment of participants from one local area in Kerala enabled us to map service provision and gain a holistic understanding of how to utilise the existing workforce to target social risk factors. The findings suggest that social interventions need to focus on ensuring health behaviour information reaches all parts of the community, and that those with more social risk factors are identified and supported to engage with treatment. Further research is required to test the resulting intervention model.

## 1. Introduction

This paper reaches towards the development of an intervention addressing the social determinants of non-communicable disease (NCD) in India. We examined a semi-urban community in Kerala, focussing on the behavioural (lifestyle), social and socio-economic risk factors relating to NCDs. The resulting intervention model examines the social determinants relating to hypertension and diabetes. We specifically targeted social factors which are amenable to modification by healthcare practitioners, social workers and community health workers working within the existing health system. We critically explored the relevance of both social capital and social support theory to the intervention model [1,2]. The resulting intervention model draws upon in-depth qualitative interviews with both community members and health care providers, and is underpinned by a Theory of Change.

### 1.1. Non-Communicable Disease in India

Diabetes Mellitus and hypertension are the most ubiquitous and important NCDs in low- and middle-income countries. Cardiovascular disease (CVD), with hypertension as its foremost risk factor, has not only the highest prevalence rate of NCDs in India, but the mortality due to CVD is rising. In 2016, 28% of all deaths in India were caused by CVD compared with 15% in 1990 [3]. Disturbingly, there are indications of rising rates of mortality due to CVD in younger Indian populations [4]. Management of NCDs in India is suboptimal. In urban populations only 38% of those with diagnosed hypertension actually receive treatment. Of those treated, only 20% have their blood pressure controlled. Equivalent figures in rural areas are 25% and 10% [5]. Another challenge is the lack of awareness of hypertension in the population: only 25% of people with hypertension in rural areas and 42% in urban areas are aware of their hypertensive status [6].

Diabetes is common in India. Following China, India has the second highest prevalence (>61 million) of people suffering from Type 2 Diabetes Mellitus worldwide, a figure predicted to double by 2030 [7]. Receiving effective treatment is imperative to avoid sequelae of both hypertension and diabetes such as stroke, cardiac failure, renal failure, peripheral vascular disease and blindness. Blood pressure control has been shown to reduce the risk for stroke by approximately one third [8]. Poverty does not confer immunity for NCDs. The rate of hypertension in the poorest households is currently almost as high as in the wealthiest households [9]. Relevant to this is the fact that the sustained, effective management of diabetes and hypertension with ongoing costs can be a significant financial burden in less affluent sections of Indian communities, leading to catastrophic financial loss and descent into abject poverty [10].

With urbanisation in India, there is evidence of increasing rates of hypertension [11], considered to be related to dietary changes and increased sedentary life. Medication is the mainstay of the management of both hypertension and diabetes, but behavioural interventions to mitigate NCDs and NCD risk factors have shown promise. A population-based strategy addressing lifestyle and societal factors in India has been shown to be effective in the reduction of not only salt intake, but also smoking and physical inactivity [12]. Moreover, NCDs such as hypertension and diabetes interact with a number of complex socioeconomic and societal variables such as poverty [9]. Recently it has been shown that the management of NCDs in India is complicated by mental health problems such as depression and anxiety as well as limited social connectedness and demographics [13]. This is in keeping with previous research (e.g., [14]).

### 1.2. Complexity of Social Determinants of NCDs

The prevalence of NCDs in Kerala is the highest in India and NCDs account for 90% of premature deaths in people between 15 and 69 years of age. This is despite government-supported medical institutions and a well-developed public health system consisting of regional clinics as well as a well-organized community health worker network [15]. Moreover, there appears to be gender differences in susceptibility to NCDs in India and the prevalence of NCDs is particularly high among women [16]. According to these authors, unmarried or separated women were at greater risk than currently married women with the highest risk found in elderly females who were either widowed or divorced. The findings are nuanced in that women who were illiterate or with poor education were found to have a higher risk for NCDs compared to men with equal educational attainments [16]. Singh et al. [17] similarly found significantly higher rates of NCDs in women in Kerala aged between of 50 and 69 years who were lonely and socially isolated, features frequently seen in India after divorce or in widowhood. In addition, abdominal obesity was significantly more prevalent in females than males in Kerala, with increased girth predisposing them to the metabolic syndrome and diabetes [15].

In a recent catchment area study in Kerala, physical, behavioural and social risk factors for CVD were explored, and it was found that there was a clustering of behavioural and social risks [18]. Although behavioural risks of higher alcohol and tobacco consumption were more prevalent in men, social risks including disintegrated social support network, low income, high life stress and low literacy levels were typically found among older women, of which a high proportion of whom were not married, and demonstrating high levels of disability, as well as depression and anxiety [18]. Despite these findings, until now, social risk pertaining to NCDs are infrequently addressed in public health interventions in India [14]. Social connections and relationships are therefore social factors of significant importance for older people in India [19].

Social networks and social support have been identified as important determinants of risk and eventually the clinical outcomes for CVD [20]. The social networks of individuals have been shown to reinforce both positive or negative behaviours in the management of chronic illness such as hypertension [21,22]. The absence of social support has long been recognised to increase the risk for CVD mortality [23], but effective interventions for low social support have remained elusive [20]. Engaging with the networks of people at risk for CVD may be an effective approach, but such interventions need to be undertaken with an understanding of the complex interplay of these diverse factors, and with full recognition of the social and cultural issues which mediate as barriers to self-management [24]. For example, Lumagbas et al. [25] described the complex interaction of social determinants such as poverty or low education with structural, behavioural and psychosocial factors as a composite matrix of factors which together increased the risk of CVD. It is likely that a unique approach that accounts for the social and cultural context will be required to address CVD risk in the Indian context.

### 1.3. Social Intervention Design Considerations

Tackling determinants of health such as a lack of education, lack of clean water, poverty, pollution and poor diet is, of course, important, though causal pathways are dynamic and multilayered [25]. For instance, it may appear that behaviour plays a larger role than poverty or lack of clean water [26]. However, stress over extended periods of time (possibly caused by poverty, which is known to be a cumulative cause of stress [27]) has been shown to influence behaviour. Behaviours linked to stress, if prolonged, have as much deleterious effect as hypertension and obesity in terms of NCD risk, and myocardial infarction is strongly related to work stress, financial stress, generalized stress and home stress, compared to controls free from these [28].

Like Lumagbas et al. [25], Byrne [29] argues that it is essential to identify key constructs and create targeted interventions with stakeholders. Clearly this indicates a move away from addressing poverty or a lack of education, but demonstrates an increasing awareness that other factors should be researched in more granularity. In Kerala, for example, a state with almost universal literacy, the high level of NCDs are not readily explained by known social determinants. Behavioural risk factors for NCDs are as prevalent in Kerala as in the USA [30], with some evidence that they are worsening [31]. The epidemiological and demographic transition in Kerala has been noted as more advanced than elsewhere in India [32], though states such as Tamil Nadu are not far behind [33]. The situation is indeed complex and a “whole system” approach is required to tackle behaviour change [26], which needs to consider both intrapersonal and interpersonal factors [34]. Intervention development needs to be theory driven, as this approach allows the identification of key constructs which inform targeted interventions, to make generalizations and synthesize findings over several contexts and thus unravel the (often) multiple components of a successful intervention [29]. Our approach has therefore been, in keeping with the recommendations of Alcántara et al. [35], to focus on integrating knowledge of social determinants of NCD risk in Kerala within a behaviour change framework. This study builds on the integrated framework of social determinants of health and the science of behaviour change to develop an intervention framework addressing social risk factors for NCDs in Kerala. We start, though, with a systematic search for effective social interventions to use existing evidence to construct our intervention framework.

### 1.4. Literature Review

To review the evidence for effective social interventions which addressed social risk factors for CVD in India, we conducted a rapid review of Medline, Scopus, PsychINFO and Embase using the search string: “((diabet* or ‘diabetic mellitus’ or ‘blood sugar’ or ‘blood glucose’ or ‘type 2 diabetes’) and (hypertens* or ‘blood pressure’ or ‘high adj3 pressure’ or ‘people adj3 hypertension’) and ‘interven*’ and (social or behaviour* or emotional or factor* or ‘risk factor’) and India)”.

The inclusion criteria were:Studies that examined effectiveness of social interventions that targeted social risk factors of Diabetes, hypertension and CVD;Randomised controlled trials;Conducted in India;Population aged over 30 years;Peer-reviewed journal articles;Written in English language.

Titles and abstracts were screened by three reviewers. To establish inter-rater reliability, all three reviewers independently reviewed all the four databases. All papers that potentially met the inclusion criteria (*n* = 61) were subjected to full text review. Two reviewers separately reviewed the full text papers and resolved disagreements by discussion as required. No papers met the inclusion criteria (Figure 1).

This lack of evidence highlights the need to develop a contextually relevant and theoretically informed intervention model for future. This paper aims to do this by drawing on evidence from local stakeholders to identify how social risk factors for NCDs in Kerala can be meaningfully addressed.

## 2. Materials and Methods

### 2.1. Design

In recent years, an array of guidance for intervention development has emerged, e.g., [36,37]. One of the most widely used, the UK Medical Research Council (MRC) guidance, describes intervention development as identifying (i) the evidence base, (ii) appropriate theory and (iii) modelling processes and outcomes [38]. Critiques of the MRC guidance emphasize shortcomings in its application within complex social systems [39], and advocate a greater role of theory in examining underlying mechanisms of change [40]; closer ties to actual ways of working and better articulation of how new research findings are sustained in practice after the stage of implementation [41]. Clarifying the definitions, concepts and relationships that explain change mechanisms can lead to interventions being more likely to succeed [42].

Hawe and colleagues [43] maintain that public health interventions should not be viewed as sets of decontextualized components, but as “events” within complex social systems. Viewing interventions in this way requires that practice is embedded in the development process, and in order that change can occur, an understanding of how the system functions and how to alter it is needed. To address these requirements, we followed the three steps of intervention development according to MRC guidelines, as well as paying attention to the complexities of the health and social systems in India. This paper reports data collected in the second stage of the intervention development process which enabled us to model the processes and outcomes of a complex social intervention [38].

With regard to the first phase of the intervention development process, the research team undertook a scoping review to first examine the existing literature of the major social risk factors which are associated with diabetes, hypertension and the comorbid conditions of depression and anxiety in India [13]. The evaluation criteria for social risk factors was an association of any strength (qualitative or quantitative) between a social variable and diabetes, hypertension and depression or anxiety. The review found that the literature on the major social risk factors is sparse, but was able to identify six themes emerging: demographic factors; economic aspects, social networks, life events, health barriers and health risk behaviours that increased the risk of these chronic conditions.

The importance of social support and social networks in the management of CVD has been further highlighted in a community survey [18]. This measured behavioural, social and other risk factors known or hypothesized to be associated with the poor management of CVD. These findings indicated the social risk subgroups that may benefit from targeted interventions to help manage CVD risks more effectively and thereby reduce morbidity. It provided the rationale to further explore the clustering of social, behavioural and demographic factors occurring within individuals, families and communities in Kerala.

We used Theory of Change (ToC) as a theoretical approach to programme design and evaluation that systematically captures contextual factors and provides clarity to the complex systems involved in the care of persons with chronic conditions in Kerala.

ToC encompasses the set of causal assumptions surrounding how an intervention will achieve its impact in a given context that can be tested using empirical data [39]. It organises the outcomes necessary to achieve impact onto a causal pathway, or ToC map. The activities or interventions required to progress from one outcome to the next are mapped onto the causal pathway [44]. All deliberate system changes are founded on a ToC. The presumption that academic theories will inherently prove superior to theories held by those with intimate knowledge of complex social systems is contradicted by the disappointing effects of many interventions based on social science theory alone [45]. Therefore, input from key actors within the system is essential in the intervention development process.

From a systems perspective, in the context of this study and setting, the ToC is not necessarily concerned about detailing a precise new intervention for Kerala’s health and social care services but may instead identify the functions of key intervention mechanisms in disrupting common patterns of system behaviour aimed at improving outcomes for persons with risk factors for CVD.

The ToC model was employed to (a) assess the capacity of the health and social care systems to integrate; (b) evaluate the feasibility of a primary care level integration model in managing chronic conditions; (c) collate results from several phases of research (scoping review [13]), community survey [18]), rapid systematic review (see above), qualitative interviews with people with risk factors for CVD and key stakeholders (see below)); and (d) to map existing community resources in the target population (see below). These steps used mixed methods and culminated in a modelling of the intervention components.

### 2.2. Qualitative Interviews

#### 2.2.1. Community Members

Semi-structured qualitative interviews were used to explore the lived experience of twenty people with risk factors for CVD. The sample was recruited from participants of the community survey [18], stratified by behavioural risk group (*n* = 7), social risk group (*n* = 6) and low-risk group (*n* = 7). Participants represented different religious and cultural groups from all socio-economic classes. Seven men and 13 women were interviewed and their ages were in proportion to survey respondents (behavioural risk group mean age = 59 years; social risk group mean age = 68 years; low risk group mean age = 51 years). Potential participants were approached by telephone and there were no refusals to participate.

The interviews explored treatment pathways; motivation for self-management; experience of health care; affordability; availability and acceptability of community resources; immediate and extended family support; experience of accessing formal and informal social support; challenges in adhering to treatment; and barriers in complying with medication. The role of facilitators in managing the barriers to treatment, such as Accredited Social Health Activists (ASHAs) and other health care providers were explored. The topic guide was created in conjunction with service providers and users in focus group discussions and was initially developed in English and later translated into the local language, Malayalam. The interview was piloted with three participants representing each risk group to establish acceptability and feasibility. Informed consent was obtained from the participants after informing them about the purpose of study, benefits and risks of participation, confidentiality and the right of the participant to withdraw at any time from the study. The interviews were conducted by postgraduate researchers trained by the research team [46]. The trusting relationship established with the participants at the time of community survey facilitated access and acceptance of the research process and enabled a safe space for participants to engage with the interviewer.

Interviews were recorded and data were transcribed verbatim and analysed thematically to explore the lived experience for participants in each risk group. Attride-Stirling’s [47] model of systemic thematic analysis involving the development of thematic networks from emerging themes was applied to “facilitate the structuring and depiction of themes” (p. 387). Data were coded based on recurrent issues within the text, resulting in a coding framework that was not preconceived but extracted from the data itself. This exhaustive list of issues was reviewed by the researchers to identify clusters of similar issues, which were arranged into “basic themes”. The issues within these basic themes were analysed to identify both strength and commonality of themes which resulted in “organising themes” and an overall “global theme”. The strength of each theme was determined by the number of times a theme emerged in the analysis.

#### 2.2.2. Health Practitioners and ASHAs

A group interview with six ASHAs and individual semi-structured interviews with six primary care doctors and two nurses were also conducted to discuss the findings of the community survey [18]. All interviews were conducted in the same Panchayat (a local administrative area) to understand the local context from the perspective of health practitioners.

ASHAs are female lay community health workers who are selected from within the community and trained to work as an interface between the community and the public health system. Being part of the local community, they are trusted within the community and aware of social and other dynamics in it. The ASHAs were interviewed to ascertain their views on barriers and potential facilitators to the management of chronic disease such as diabetes and hypertension in the community.

The health professionals and ASHAs were purposively selected for the interviews as they had specialist knowledge and experience of the local context of the management of risk factors for CVD. The interviews were recorded and transcribed for analysis. As these were typically briefer than the interviews with community members and yielded less data, they were analysed using thematic analysis [48] to identify the salient themes in relation to our research questions.

### 2.3. Expert Workshop

Finally, we convened an expert workshop to bring together five key local stakeholders including practitioners and policy makers. The workshop was used to review previously collected data; discuss barriers in treatment and care, workforce constraints, family structures and social factors; and develop an asset map of the local resources available to people with chronic health conditions in the study site. Data from the qualitative interviews and workshop were synthesised to inform the development of the ToC for the complex social intervention.

### 2.4. Ethical Approval

Ethical approval for the study was obtained from the Rajagiri Hospital Institutional Ethics Committee (ref. RAJH18003). Full written informed consent to participate in the study was obtained from each participant.

## 3. Results

### 3.1. Qualitative Interviews

#### 3.1.1. Community Members

An exhaustive list of basic themes emerged from the data analysis with 29 being selected. Sub-issues within these basic themes emerged and were rearranged into organising themes with an overall global theme of “Social risk and chronic disease management” (Table 1). The data was coded in relation to the three risk groups (group 1 = low risk; group 2 = behavioural risk; group 3 = social risk). Interestingly the same organising themes emerged for each group and provided the basic framework for the thematic analysis.

The low-risk group (Figure 2) did not appear aware that lifestyle factors could increase their chronic disease risk, though they seemed to understand the importance of a healthy diet. That they were financially stable meant that they had better choices and were able to access higher quality services. Where they experienced positive friendship and family support, they appeared to be motivated to adhere to their medication regime and to self-manage their diet and exercise programme. However, where they were socially isolated, they lacked the interpersonal support to motivate them to access readily available resources. One of the most significant benefits for this group was their capacity to be able to engage in decision making with the freedom to choose quality healthcare. This included regular monitoring with medication and lifestyle advice in a positive healthcare setting. If needed, they could access counselling services and insurance which helped them to deal with emergency situations. In general, their economic security and stability resulted in their accessing appropriate and meaningful services.

The behavioural risk group (Figure 3) were characterised by multiple barriers which related to habits and physical factors. Their diets generally contained an excess of “non-healthy” foods, in particular a high level of sugar intake. Their eating habits were irregular and frequently erratic. This was linked to workplace constraints and long hours which made it challenging to follow a healthy diet or to keep to treatment programmes. They found it difficult to access clinics for medication and to follow advice to change their lifestyle. Awareness that was gained either through personal experience or witnessing a change in others’ lives had a positive effect on overall treatment adherence. Where people were employed, they were able to afford ongoing medication, diet and exercise. However, where there was a lack of stable income the needs of the family had to come first over medication costs. A supportive family was found to be powerful in accessing resources despite long distances to clinics, and they reminded and encouraged medication adherence. However, many opted for alternative treatments such as Ayurveda or homoeopathy because they were not convinced that medical professionals knew best and obtained good results. There was a lack of trust in government doctors and fears of becoming addicted to long term medication.

The *social risk group* (Figure 4) was characterised by social isolation, financial distress and a lack of meaningful social interpersonal relationships. Their isolation from health services on a routine basis resulted in poor understanding of preventive practices leading to a reduction in risk for chronic disease. Religion was a source of restoring hope. ASHAs were acknowledged as important sources of information about the importance of lifestyle change through frontline community health programs. Financial distress was critical in determining lifestyle choices and access to available resources. Where there was financial distress, people found it difficult to afford a healthy diet or buy medicines. Many were receiving help from churches, mosques or community philanthropists to help them buy medication and pay for their travel expenses to community health clinics. Where there was family support, this reduced social and community isolation and resources were more readily accessed. The group was characterised by insufficient finances to afford private health care and described difficulties in travelling to government hospitals which were frequently long distances away from their homes and overcrowded.

#### 3.1.2. Health Practitioners and ASHAs

Three prominent themes emerged from the interviews with primary care doctors and nurses. Firstly, they noted a poor public awareness of the importance of adherence to medication to manage the risks for CVD and the misconception that privately purchased medication was more efficacious than government subsidized medications provided at primary health centres (PHC). They also discussed the lack of public awareness of government services such as NCD clinics.

Secondly, the doctors highlighted that task-shifting could potentially improve the management of risk factors for CVD. They were aware of overcrowding and rushed consultations at medical clinics and recommended that medication such as insulin could be supplied from PHCs, thereby alleviating the burden on specialist medical clinics. In addition, they felt that ASHAs should be trained in metrics such as finger-prick blood glucose testing and blood pressure measurements and that there should be regular upskilling of ASHAs in the effective management of NCDs. They suggested that ASHAs could be trained in counselling or provide medication to those unable to afford it. They noted that there was potential to employ social workers at PHCs who could assist in educating the public in managing risk factors for CVDs.

Thirdly, the doctors felt that there was a dearth of family and other support for people with chronic disease and that psychological support was often unavailable. They were aware of barriers to the management of CVD risk factors, largely in poorer sections of the population related to the cost of transport to clinics. In particular, people should be supported to obtain and take medication, and be made aware of the consequences of non-compliance and the need for a healthy diet and adequate exercise.

ASHAs expressed similar views to the doctors. They expressed interest and willingness to be upskilled in the management of chronic disease. They noted that many were involved in record-keeping for maternal and child health and providing support with palliative care. However, because of the high and rising burden of NCDs such as hypertension and diabetes they were unanimous in their interest and desire to be involved in the management of these conditions at community level. They discussed how they reported to the nursing staff at the PHCs on a monthly basis and felt that management of chronic disease could be incorporated into their roles.

The ASHAs noted a number of potential facilitators to their involvement in the management of risk factors for CVD: all had and used mobile phones; all valued the feedback to the PHC; the appreciation from the community for their efforts was significant; there was a NCD clinic functioning locally for three wards. However, they also identified three barriers to overcome. Firstly, they acknowledged the need for training relating to diabetes and hypertension; the use of finger-prick glucose measurement or a sphygmomanometer; common anti-diabetic and anti-hypertensive medications and their side-effects; and what action to take under different circumstances, including now best to support people with mental health problems. Secondly, they acknowledged barriers in access to medication as there were often shortages at both ward and sub-centre levels, and medication brands were frequently changed leading to confusion and suspicion among people prescribed them. Some people also had to travel considerable distances to PHCs to obtain them, which was prohibitively expensive. Thirdly, they expressed a lack of immediate support for their role and felt isolated in the management of people with risks for CVD.

### 3.2. Stakeholder Workshop

A stakeholder workshop was held in October 2019 with five stakeholders including primary care physicians, junior health inspectors and other health care specialists from community medicine and psychiatry. The theory of change workshop aimed to share perspectives on priority areas of intervention in managing risks for CVD in Kerala, clearly define the problems within the current treatment models and healthcare context and begin to determine which specific mechanisms for change may be feasible in the population and resources needed. Themes from the discussion are summarized in Table 2.

### 3.3. Mapping Resources in the Community

A resource mapping exercise was completed in one district to identify and classify the existing resources through discussions with community leaders and government public health staff. Figure A1 in Appendix A illustrates the public sector service across four themes of physical health, psychological, social and economic support. Several resources (e.g., ASHAs, Family Care Centres) potentially provide multiple types of support including medical, psychological and social care to people with risks for CVD.

Additionally, the following private sector services were available in the district:*Health*: religious organisations provide nursing care services for older people, home care for those unable to reach health facilities, postnatal maternal and infant care; NGO’s provide regular medical camps, community-based palliative care and free medications based on need.*Psychological and social support*: at the community level, friends, neighbours and family members provide much social support.*Economic*: private organizations provide medication, medical kits, and a monetary support to the caregivers of people who require home assistance. Financial support is also provided to some in need through religious organisations.

## 4. Discussion

Social constructs such as social capital, social networks and social support may a priori inform an effective intervention design as they address the social context of people lives and factors which may compound their risk for NCD. The network approach to social capital [49] may be important as it enables us to explore the dynamic interaction of power structures and the exchange of resources within families, networks and local communities. Understanding the dynamics of social capital may help to explain why traditional public health interventions to reduce the risk of NCD may not either reach or be effective in some population sub-groups. For example, older women with high levels of disability and limited support networks and who are at higher risk for CVD [18], may have limited access to resources within their family or immediate network [49], and thus impair their ability to access treatment for diabetes or hypertension. Additionally, as people with lower literacy levels rely more on conversational (verbal) communication for their health information [50], it is important that social interventions are directed towards those who are isolated from their local community networks.

Social disadvantage, poverty and unemployment are factors which significantly increase the risk for NCDs. However, social determinants do not have an homogenous effect on health [35], indicating that a more nuanced approach to the management of social risk for CVD is required. This includes, for example, an understanding of different family systems and relationships which provide care for people and facilitate (or impede) access to community or health resources. In the relatively wealthy state of Kerala, with high levels of education and low levels of absolute poverty, social risk factors impede individuals’ access to community or healthcare which impact on their ability to manage risk factors for CVD. Our study suggests that these should be the target of our ToC model.

Data collected for this study revealed significant concordance between the interviews of doctors, ASHAs, community members and the findings of the community survey. Those in the social risk group identified a lack of support from family, relatives and neighbours, and hopelessness about the future due to a lack of support from their children. Many felt isolated due to a lack of interaction with their neighbours. The cost of travel prevented them from accessing PHCs, purchasing medications, doing tests and changing their diets. The high prevalence of depression and anxiety in the “social risk group” [18] was reflected in the doctors’ comments about their need for psychological support. The lower levels of income, education and employment, older age, higher levels of disability and less integrated social networks found in the social risk group [18] were also reflected in the findings of the qualitative interviews (Figure 3).

These findings suggest that it is more than individual social risk factors which need to be addressed. Risk factors appeared to be clustered and interacting, requiring complex and multi-faceted responses. This supports previous findings that contextual characteristics of individuals and communities in India impact on NCDs [17] and that a multisectoral approach is required to address them [51]. For example, poverty is not purely about a lack of resources. While those with money could choose private care, even those lacking financial resources would prefer to pay for private care rather than accept free medication because of beliefs about its inferiority. A more inclusive, holistic and nuanced approach to the management of NCD risk is required for those experiencing a greater preponderance of social, economic and family problems. For a complex social intervention to be successful in India, it needs to account for the resources in families and communities, and the social context of both.

Interventions focusing on social capital have shown promise in improving the health of older people, though approaches have been diverse and the findings mixed [52]. The relationship of social capital and diabetes is as yet unclear as only a few studies have been conducted and measurement of the concept is not robust [53]. Health research is not alone in adopting the concept uncritically [54]. There is therefore a need to give careful and critical consideration to social capital in the design and evaluation of interventions to ensure they are successful [2]. In particular, “linking” social capital, which emphasises the importance of connecting marginalised people to formal institutions [55], needs to inform intervention design to help ensure that people are connected to the health resources they require. Primary care services and social welfare provision need to be connected to help ensure that social risks for CVD are managed more effectively. Our findings indicate that services need to be better connected and the intervention ToC needs to focus more on systems as a whole.

A further feature of the social risk group that requires attention is social isolation and a lack of connection with the local community, though there is some evidence that people within this group gain informal support from churches or mosques (Table 1). Interventions which enhance weak ties in people’s networks may be required, in order to connect them to health resources and to public health communications. Granovetter’s [56] “strength of weak ties” hypothesis suggests that people benefit from information shared by informal contacts. It has become foundational in social network scholarship because weak ties, in contrast to strong ties such as close family members who may provide emotional warmth, can also bring benefits such as enhanced community connections or opportunities beyond the household. While emotional support is known to reduce the risk for CVD [57] and is therefore important, it is perhaps equally important to ensure that people have access to public health information and networks of people who can support behaviour change.

Our literature review found no social interventions in India with evidence of their effectiveness to inform our ToC model. However, interventions evaluated elsewhere in South Asia, such as motivational interviewing [58], or international evidence of the importance of social support to the efficacy of dietary and physical activity interventions [59], were used to inform the intervention development process.

To model intervention processes and outcomes for this group, after consideration of relevant theory and evidence, we combined results from the previous research and the findings presented in this paper with the knowledge of the health and social systems, as pertaining to the community studied, by members of the research team to propose a new intervention model. Several key features of the ToC emerged.

Firstly, ASHAs are ideally positioned to respond to the issues which emerged from our findings, as they know their local community and are trusted by it. However, we acknowledge their heavy workload and it may not be possible to add the management of social risk factors of NCDs to the list of their responsibilities. Where possible, though, training could be provided to increase their capacity and capability to undertake this role.

Secondly, in view of the fact that there are no social workers employed in the community in public health positions (while this is true for Kerala we acknowledge it is unknown for all states in India), we suggest that those engaged in welfare organisations in the community could be provided with enhanced training in CVD risk factor management which they could integrate into their work. Social workers in India already receive training in a “medical and psychiatric social work” specialisation so there is a trained workforce equipped with many of the required skills and knowledge for this role.

Thirdly, primary health care and social welfare systems of service delivery need to be better connected to ensure people at highest risk of CVD are identified and supported appropriately. Both ASHAs and social workers will require training in screening tools to identify social risk factors such as social isolation, poverty, family relationship issues and chronic health conditions in local populations. Referral systems need to be established to ensure those with social risk indicators and chronic health conditions are connected to specially trained social workers.

Finally, social work intervention would comprise linkage to existing resources within communities and access to local health services; addressing social needs and providing behavioural health interventions; and providing case management with other health providers.

By developing the ToC, we will be able to determine the potential impact of the intervention on treating and managing the risk for CVD and the behavioural and social risk factors that may have an effect on the desired outcomes. This helps to understand (and visualise) the contextual features that may impede or facilitate the implementation of an integrated care model in India.

### Limitations

This study has been conducted in one panchayat in Kerala. While this has provided us with some rich insights into the needs of the local population and the health services available to meet them, it reduces the generalisability of this study. An intervention model developed on the basis of these findings will need to be tested in several locations to establish its potential for widespread application throughout India. Indeed, the theoretical propositions considered here need to be contextualised in future study locations to assess their relevance to local needs and to ensure that interventions are sensitive to local social and health systems.

This study does not claim to be representative of all communities in India and its small sample size limits the inferences which should be drawn. However, it is noteworthy that the findings from the qualitative interviews with community members, stratified by risk group, lent validity to the findings of the latent class analysis conducted in previous research [18] which first identified the distinct clusters of risk factors. Further research in larger samples is required to further validate these findings.

## 5. Conclusions

Social determinants of NCDs in India are complex and interacting. Single or universal public health approaches to address them are unlikely to be effective as they do not adequately address clusters of social risk factors. An approach which integrates health and social welfare systems appears to be needed, which engages with social systems within and beyond families. This study found a system of services and support for people to manage risk factors for NCDs already exists in Kerala, but some co-ordination is required to bring it together to target resources towards those whose social risk factors are not currently being addressed.

## Figures and Tables

**Figure 1 ijerph-17-08636-f001:**
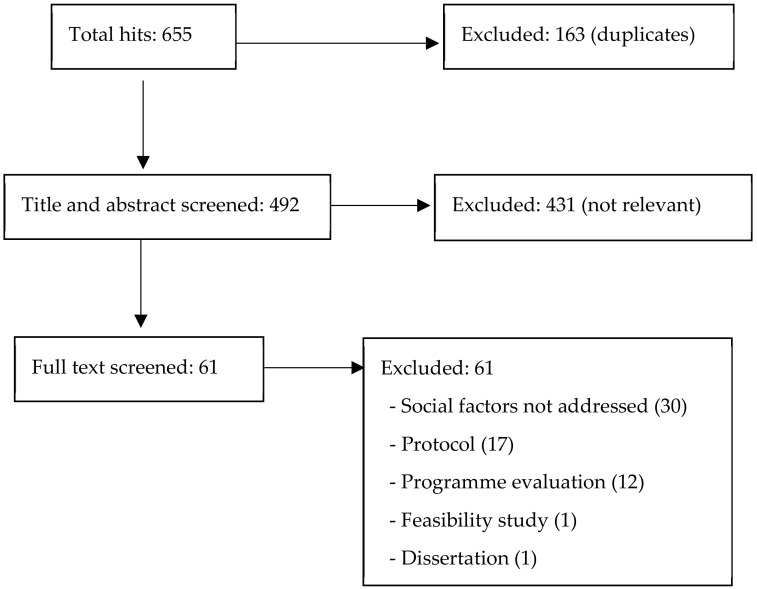
Literature review flow diagram.

**Figure 2 ijerph-17-08636-f002:**
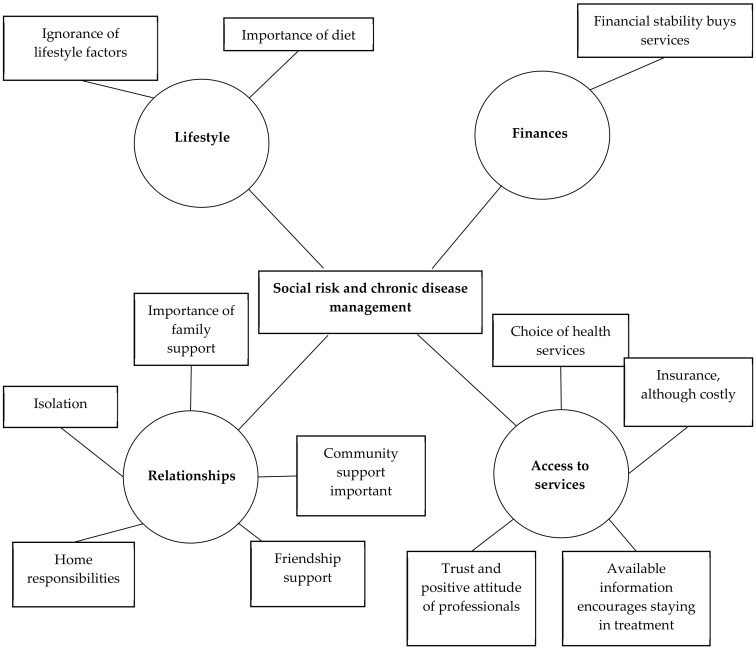
Thematic network: Low-Risk Group.

**Figure 3 ijerph-17-08636-f003:**
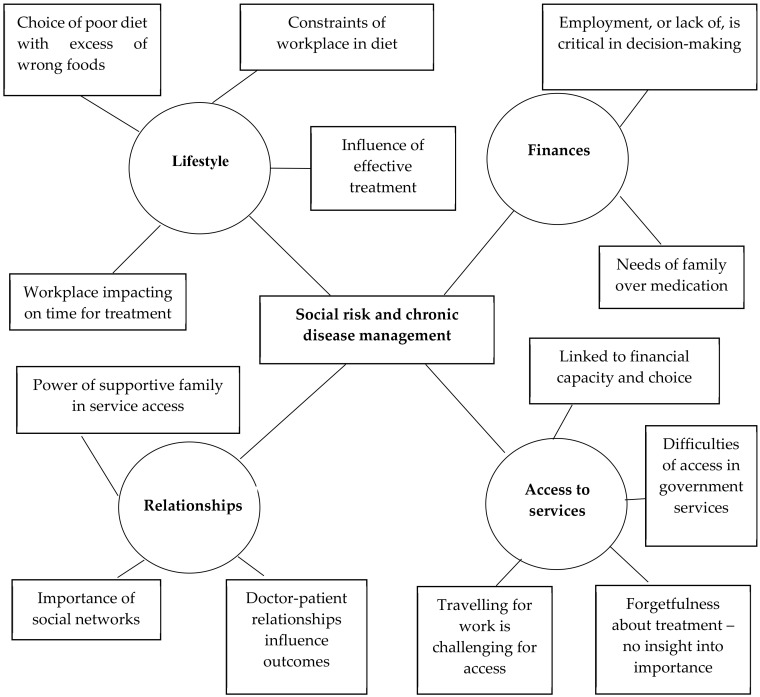
Thematic network: Behavioural Risk Group.

**Figure 4 ijerph-17-08636-f004:**
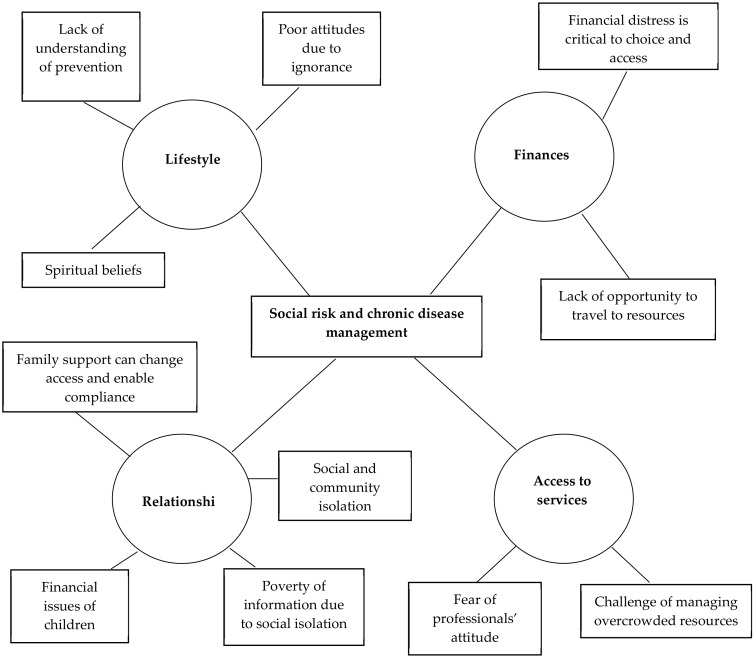
Thematic network: Social Risk Group.

**Table 1 ijerph-17-08636-t001:** Summary of themes from community members.

Basic Themes	Organising Themes	Global Theme
	Group 1—Low Risk	Social Risk and Chronic Disease Management
Individuals are unaware of the health risks and complications Incorrect diet and lack of awareness	Lifestyle	
Financial Stability opens way for better choices and accessing quality services	Finance	
Lack of immediate family support to buy medicines Community support through health education Support of friends in the form of checking in, buying medicines, etc.	Relationships	
Trusting, friendly and positive attitude of healthcare professionals Economic stability results in accessing better quality services and informal support	Access to services	
	**Group 2—Behavioural Risk**	
Job and busy routine leading to non-adherence to following healthy diet Lack of education or knowledge leads to ignorance which, in turn, affects adherence	Lifestyle	
Unemployment with lack of stable income Choosing to meet the needs of family over buying medicines for self	Finance	
Physical and emotional support from immediate family members Networking by extended family members in the form of referrals and community linkages Positive doctor–patient relationship and involvement of health care professionals	Relationships	
Opting for alternative treatments (Ayurveda/homeopathy/naturopathy) Lack of trust towards government doctors Wrong notions such as one gets addicted to medicines together with forgetfulness leads to non-adherence Frequent travel as a barrier to non-adherence	Access to services	
	**Group 3—Social Risk**	
Awareness about the significance of the illness and complications resulting from non-adherence Spiritual beliefs and ventilation as a source of coping mechanism and restoring hope	Lifestyle	
Lack of finances as a reason to settle for government hospitals over private ones Lack of finances leading to inability to follow healthy, balanced diet and to buy medicines Informal support from churches or mosques	Finance	
Belief that own children won’t provide support because of financial constraints or conflicting relationships Life events like death of spouse Lack of neighbours and social isolation	Relationships	
Lack of transport to reach long distance resources Rude approach from healthcare professionals	Access to services	

**Table 2 ijerph-17-08636-t002:** Stakeholder workshop results.

Challenges	Needs
Health care providers focus on medical treatment with little time or resources for care The community workforce is grassroots, reaching those in need, but extremely overloaded Assessment and intervention with social relationships a priority of family agencies employing social workers, however links between these agencies and health care providers is not always strong Systematic challenges: little time and resources for care Many government services are being provided but some lack awareness in the community, issues of accessibility for vulnerable populations Challenge for government services is the myth that free medicines or treatment are inadequate	*Health care context* Move from curative to preventive interventions Increased access to already existing mental health services like the District Mental Health Programme. Health volunteers exist for mental and neurological care, potential for other chronic conditions
	*Community care* Strengthen lower-cost parts of this system, leaving clinical as it is but increasing access to ASHAs and social workers. Systems where patients are able get their medicines at their home Systems for coordination across workforce and services
	*Family support and self-care* Generate feasible systems through existing self-help groups to enhance social and emotional support Links are needed between community workers and supportive community programmes such as Kudumbasree, a trusted system of neighbourhood groups for poverty alleviation
	*Awareness raising* Sensitization about the services that are currently available (government services like free medicines, NCD clinics, etc.) Awareness of medication compliance is lacking

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
