# Peer review of "Towards the Development of an Intervention to Address Social Determinants of Non-Communicable Disease in Kerala, India: A Mixed Methods Study"

_ijerph, 2020, doi:10.3390/ijerph17228636_

Round 1
Reviewer 1 Report
This is a well written paper outlining the development of an intervention to address social determinants of non-communicable disease in India. The authors have interviewed with CVD (n=20), Accredited Social Health Activists (n=6) and health professionals (n=8). Although the idea is well needed in the current scenario of NCD's in India, the sample size is far too small to draw solid conclusions. I would highly recommend increasing the sample size. However, this has been mentioned by the authors in the limitations section.
Major revisions needed for
1. Other papers that explore NCDs in India need to be significantly discussed and addressed in this paper, especially in the discussion section.
Reviewer 2 Report
This is a well-researched paper, dealing with a research topic of interest within the frame of public health as the social determinants of non-communicable diseases as diabetes and hypertension in the Indian region of Kerala.
The manuscript is well-written and the authors proof to know the main literature in this topic and the selected methodology is the most appropriate for their research purposes.
I have just a couple of suggestions that could contribute to the improvement of the final manuscript:
- The authors recognize the difficulty to generalize their findings to the whole country due to the specificities of Kerala, the region analyzed in this manuscript. Therefore, the word "India" must be removed from the title as the authors do not provide any insight about the possibility to extrapolate their findings to the whole country. Anyway, if the authors insist to keep this word in the title, they must provide at least a comparison of the profile of Kerala and India in terms of the six major six social risks pointed in the manuscript: demographic factors; economic aspects, social networks, life events, health barriers and health risk behaviours that increased the risk of these chronic conditions.
- The authors state in line 108 " For instance, there are indications that behaviour may play a larger role than poverty or lack of clean water. Marmot and Bell [26] argue that the mind itself, with cognitions, emotions and attitudes, is an important substrate for NCD risk. Ongoing stress over extended periods of time has been shown to influence behaviour. Behaviours linked to stress, if prolonged, have as much deleterious effect as hypertension and obesity in terms of NCD risk, and myocardial infarction is strongly related to work-stress, financial stress, generalized stress and home stress, compared to controls free from these [27]". This is unclear as it is not stated the existing link between poverty at different life-course stages and stress as a result of cumulative risk exposure. In other words, poverty is one of the main sources of stress over life. Consequently, behaviors are usually conditioned by poverty and we cannot say that one is more important than the other (see: Evans, G. W., & Kim, P. (2007). Childhood poverty and health: Cumulative risk exposure and stress dysregulation. Psychological Science, 18(11), 953-957.)
Reviewer 3 Report
Lines 21-23 - For 'qualitative interviews with people at risk of CVD (n=20)', does this mean that the Accredited Social Health Activists (n=6), health professionals (n=8), and a stakeholder workshop (n=5 participants) were not at risk of CVD?
Lines 147-149- The figure is very insightful to demonstrate a need for this study.
Lines 212-215- Is this survey instrument or a sample of the questions asked available?
Lines 263- Does 'community members' refer to Health practitioners and ASHAs or the expert workshop? Or both?
This study is insightful and pertinent. Clarification of methodological processes can enhance the paper.
Reviewer 4 Report
Thank you for letting me review this study of Social Significance.
It is a fascinating and valuable action, and I am looking forward to future studies on improving India's health care with the 'new intervention model' presented in the discussion.
Here are my concerns:
- Literature review
In the literature review on 'Effective Social Interventions', even if there are no studies the same as this, but there are some points/parts of other studies that can be referred to. In Japan, South Korea, or Taiwan, some social interventions are conducted, and could be referred to about the methods and effects of social interventions.
I think it is an overstatement to label 'Randomised controlled trials' and 'Conducted in India' in inclusion criteria.
- Materials and Methods
Please specify the evaluation criteria for risk factors for CVD.
"A mixed methods study" on the title, please mention it in the Method.
3. Please explain the definition of Social Capital used in this paper. Depending on studies, Social networks and Social support may also be included in Social Capital.
